# Effects of Prospective Audit and Feedback in Patients with Extended-Spectrum *β*-Lactamase-Producing *Escherichia coli* Bacteremia

**DOI:** 10.3390/microorganisms12112275

**Published:** 2024-11-09

**Authors:** Yota Yamada, Motoyasu Miyazaki, Hisako Kushima, Hitomi Hirata, Arata Ogawa, Yukie Komiya, Chika Hagiwara, Akio Nakashima, Hiroshi Ishii, Osamu Imakyure

**Affiliations:** 1Department of Pharmacy, Fukuoka University Chikushi Hospital, Fukuoka 818-8502, Japan; motoyasu@fukuoka-u.ac.jp (M.M.); h.hirata.cd@adm.fukuoka-u.ac.jp (H.H.); a.ogawa.ch@adm.fukuoka-u.ac.jp (A.O.); kubota114@fukuoka-u.ac.jp (C.H.); anakashima@fukuoka-u.ac.jp (A.N.); imakyure@fukuoka-u.ac.jp (O.I.); 2Department of Infection Control and Prevention, Fukuoka University Chikushi Hospital, Fukuoka 818-8502, Japan; hkushi@fukuoka-u.ac.jp (H.K.); yukiek@adm.fukuoka-u.ac.jp (Y.K.); hishii@fukuoka-u.ac.jp (H.I.); 3Department of Hospital Pharmacy, Faculty of Pharmaceutical Sciences, Fukuoka University, Fukuoka 814-0180, Japan; 4Department of Respiratory Medicine, Fukuoka University Chikushi Hospital, Fukuoka 818-8502, Japan; 5Department of Clinical Laboratory, Fukuoka University Chikushi Hospital, Fukuoka 818-8502, Japan

**Keywords:** antimicrobial stewardship, prospective audit and feedback, extended-spectrum *β*-lactamase-producing *Escherichia coli*, treatment failure rate

## Abstract

Antimicrobial stewardship (AS) Guidelines by the Infectious Diseases Society of America recommend employing prospective audit and feedback (PAF) as an effective intervention in AS programs. Since July 2022, our hospital has implemented PAF for all patients with positive blood cultures, including those with extended-spectrum *β*-lactamase (ESBL)-producing *Escherichia coli* (EC) bacteremia. Our study examined the effect of PAF on clinical outcomes in patients with ESBL-EC bacteremia. We enrolled 62 patients diagnosed with ESBL-EC via blood culture who were undergoing antibiotic treatment. The patients were divided into the pre-PAF and post-PAF implementation groups. The rate of antibiotic de-escalation from broad-spectrum antibiotics to narrow-spectrum cefmetazole was significantly higher in the post-PAF group than in the pre-PAF group (80.7% vs. 32.4%, *p* = 0.0003). The treatment failure rate in the pre-PAF group was higher than that in the post-PAF group (38.7% vs. 12.9%, *p* = 0.04). The results of this study indicate that the implementation of PAF is advantageous not only in terms of process indicators but also in improved clinical outcomes, including reduced treatment failure rates. We hope that this study will encourage the implementation of PAF in more facilities to instigate a collective effort to reduce the incidence of antimicrobial resistance.

## 1. Introduction

The emergence of antimicrobial resistance (AMR) bacteria, such as multidrug-resistant *Pseudomonas aeruginosa* (MDRP) and extended-spectrum *β*-lactamase (ESBL)-producing bacteria, is increasing worldwide [1,2,3]. In 2019, an estimated 4.95 million deaths were associated with bacterial AMR, including 1.27 million deaths attributable to bacterial AMR [4]. If no interventions are implemented against AMR, more than 10 million people will die annually from AMR by 2050 [5,6]. In particular, ESBL-producing bacteria are common sources of AMR in animals and humans [7]. Infections caused by AMR *Enterobacteriaceae*, particularly ESBL-producing *Escherichia coli* (ESBL-EC), have increased worldwide and have become a serious public health concern [8]. ESBL-ECs are resistant to many *β*-lactam antibiotics, including penicillins, aztreonam, and most cephalosporins [9]. *E. coli* is a very common causative agent of bacterial infections and is the most common Gram-negative bacterial species that carries and is a significant repository of ESBL genes [10], which confer resistance to a range of *β*-lactam antibiotics [11,12,13]. ESBL-EC infections have been associated with inappropriate empirical antibiotic therapy, longer hospital stays, poorer prognosis, and higher healthcare costs [14,15,16].

Antimicrobial stewardship (AS) can mitigate the emergence of bacterial resistance and improve clinical outcomes [17]. The Infectious Diseases Society of America and the Society for Healthcare Epidemiology of America guidelines recommend the implementation of AS by an AS team (AST), which could include an infectious disease specialist, pharmacist, microbiologist, and nurse, to facilitate the treatment of infectious diseases [18]. Furthermore, the AS guidelines recommend two effective interventions: prospective audit and feedback (PAF) of infectious disease care and preauthorization of antimicrobial use [19]. PAF is one of the most important interventions for AS that involves optimizing antimicrobial therapy throughout the treatment course by prospectively monitoring infectious disease treatment and providing feedback as necessary [20].

In April 2020, Fukuoka University Chikushi Hospital launched a comprehensive AS program. At that time, the AST pharmacist determined whether the antibiotics administered upon positive blood cultures were appropriate; however, intervention was insufficient in stopping treatment, resubmission of cultures, and duration of treatment. Since July 2022, PAF has been implemented for all patients with positive blood cultures, including those with ESBL-EC bacteremia. Although several studies have examined the impact of PAF on clinical outcomes in patients with bacteremia [21,22], no studies have investigated the impact of PAF on clinical outcomes specifically in patients with ESBL-EC bacteremia. This study examined the effects of PAF on the clinical outcomes of patients with ESBL-EC bacteremia at our hospital.

## 2. Materials and Methods

### 2.1. Study Population

This retrospective cohort study was conducted at Fukuoka University Chikushi Hospital in Fukuoka, Japan, between April 2020 and March 2024. The study protocol was approved by the Ethics Committee of the Fukuoka University School of Medicine (No. C23-06-004). Patients diagnosed with ESBL-EC via blood culture who were undergoing antibiotics treatment were included in the study. For patients with ESBL-EC bacteremia, only the first episode was included in the analysis. Patients who were transferred to another hospital during treatment without improvement were excluded from the study.

### 2.2. Microbiological Analysis

Blood cultures were processed using the FilmArray^®^ (BioFire Diagnostics LLC, Salt Lake City, UT, USA) system. The Film Array^®^ blood culture identification panel (BCID), a highly multiplexed polymerase chain reaction assay, can identify 24 etiologic agents of sepsis (8 Gram-positive bacteria, 11 Gram-negative bacteria, and 5 yeast species) and 3 AMR genes (*mecA*, *vanA/B*, and *blaKPC*) from positive blood culture bottles. The Film Array^®^ blood culture identification 2 panel (BCID2) is a molecular diagnostic tool for rapidly identifying pathogens and resistance genes directly from positive blood cultures. The updated version of the original BCID, designated as BCID2, identified 33 species, including a new distinction between *Enterococcus faecalis* and *E. faecium*. Moreover, it examined 10 genetic resistance markers, including prevalent carbapenemase-encoding genes (e.g., *blaKPC*, *blaIMP*, *blaNDM*, *blaOXA-48-like*, and *blaVIM*); the most common ESBL gene (*blaCTX-M*), and a genetic marker for colistin resistance (mcr-1).

### 2.3. PAF Intervention

The bacteriology laboratory at our hospital is required to report positive blood cultures to the AST from Monday to Friday, excluding public holidays. The AST confirmed the type of infectious disease and antibiotics from the medical records and immediately intervened in the event of an incompatibility between the assumed or identified bacteria and the antibiotics. Once the susceptibility results are known, the AST will recommend a change in the antibiotic treatment to the attending physician, including de-escalation, while considering the course of treatment. If necessary, the time at which the antibiotics should be terminated is proposed. Our hospital introduced BCID in April 2020, which coincided with the commencement of activities associated with the AS. In December 2022, the hospital introduced a new diagnostic tool, BCID2, to facilitate the early diagnosis and treatment of patients presenting with bacteremia. After BCID2 introduction, if ESBL-EC is detected in blood cultures and ESBL-EC is not susceptible to the initial treatment, switching to an antibiotic generally effective against ESBL-EC, such as cefmetazole (CMZ) or meropenem (MEPM), is recommended depending on the severity of the disease.

### 2.4. Clinical Characteristics

The patients’ clinical data were collected from electronic medical records and retrospectively evaluated. Patient characteristics included age, sex, use of anticancer and immunosuppressive agents, history of catheter insertion, surgical history, laboratory findings [white blood cell counts [WBC], C-reactive protein [CRP], and creatinine], and the detection of multiple bacteria in blood cultures. Sources of bacteremia and initial therapeutic antibiotics for patients with ESBL-EC bacteremia were investigated as well. The Charlson comorbidity index (CCI) was used to measure comorbidity [23]. The severity of infectious diseases was assessed using the Pitt bacteremia score (PBS) [24] to predict the risk of mortality in patients with bacteremia using a scale from 0 to 14 points. A score ≥ 4 is commonly used as an indicator of critical illness and increased risk of mortality [25].

### 2.5. Process Indicators

The process indicators were assessed for the following items: testing of blood cultures using BCID2, the number of days from blood culture submission to a positive blood culture result, de-escalation from broad-spectrum antibiotics to cefmetazole, the length of antibiotic treatment, the length of treatment with broad-spectrum antibiotics, and the switching from intravenous to oral antibiotics. In this study, broad-spectrum antibiotics referred to sulbactam/cefoperazone, ceftriaxone (CTRX), MEPM, biapenem, tazobactam/piperacillin, and tazobactam/ceftorozan.

### 2.6. Clinical Outcomes

Clinical outcome was assessed by the length of hospital stay and treatment failure rate. Treatment failure was defined as a relapse of infections after the completion of treatment or death within 30 days of bacteremia onset.

### 2.7. Statistical Analysis

Data were expressed as the number (percentage) for categorical variables and the median (interquartile range) for continuous variables. The patients with ESBL-EC bacteremia were divided by the AST into the pre-PAF and post-PAF groups. The chi-square test or Fisher’s exact test (as appropriate) was used to compare categorical variables, and the Mann–Whitney U-test was used to compare continuous variables. Multivariate logistic regression analysis was performed for factors associated with clinical cure in univariate analysis (*p* < 0.1). Statistical significance was set at *p* < 0.05. All statistical analyses were performed using JMP^®^ 10 (SAS Institute Inc., Cary, NC, USA).

## 3. Results

### 3.1. Patient Characteristics

Of the 205 patients with *E. coli* detected by blood culture, 62 with ESBL-EC were enrolled (Figure 1). The median age of the 62 patients was 84 years (interquartile range: 77.8–90 years); 28 (45.2%) were male, 40 (64.5%) had a history of catheter insertion, 11 (17.7%) had received anticancer and immunosuppressive agents, and 16 (25.8%) had surgical histories. The median CCI was 3 (1–4), with a maximum score of 11. The median PBS was 2 (0–3), with a maximum score of 10. The laboratory findings were as follows: WBC 10,300 (6150–13,300)/µL, CRP 10 (4.9–16.4) mg/dL, and creatinine 1.06 (0.66–1.53) mg/dL. Seven (11.3%) patients had multiple bacteria detected in blood cultures. The sources of ESBL-EC bacteremia were as follows: urinary tract infection, 33 cases (53.2%); biliary tract infection, 18 cases (29.0%); respiratory tract infection, 7 cases (11.2%); catheter-related bloodstream infection, 1 case (1.6%); skin and soft tissue infection, 1 case (1.6%); pancreatic infection, 1 case (1.6%); and unknown, 1 case (1.6%). The antibiotics used in the initial treatment of patients with ESBL-EC bacteremia were as follows: sulbactam/cefoperazone, 15 cases (24.2%); CTRX, 15 cases (24.2%); MEPM, 15 cases (24.2%); tazobactam/piperacillin, 6 cases (9.7%); CMZ, 5 cases (8.1%); cefotiam, 2 cases (3.2%); sulbactam/ampicillin, 2 cases (3.2%); and biapenem, 2 cases (3.2%).

Patient characteristics at pre- and post-PAF are summarized in Table 1. No significant differences were observed in PBS and CCI between the pre- and post-PAF groups. The sources of bacteremia were also similar between the two groups. No difference in the initial usage rate of non-susceptible antibiotics against ESBL-EC, such as cefotiam and CTRX, was found between the pre- and post-PAF groups (29.0% vs. 25.8%). Similarly, the initial usage rate of carbapenems did not differ between the two groups (25.8% vs. 29.0%). Overall, no significant differences in patient characteristics were observed between the two groups.

### 3.2. Comparison of Process Indicators Between the Pre-PAF and Post-PAF Groups

No significant differences were observed between the pre-PAF and post-PAF groups in process indicators, including the number of days from blood culture submission to when a positive blood culture result was obtained, the length of antibiotics treatment, length of treatment with broad-spectrum antibiotics, and switching from intravenous to oral antibiotics (Table 2). The rate of BCID2 use was significantly higher in the post-PAF group than in the pre-PAF group (*p* < 0.0001). The rate of de-escalation from broad-spectrum antibiotics to narrow-spectrum antibiotics was also significantly higher in the post-PAF group than in the pre-PAF group (*p* = 0.0003). It should be noted that all of the narrow-spectrum antibiotics used in the de-escalation cases mentioned above were CMZ.

### 3.3. Comparison of Clinical Outcomes Between the Pre-PAF and Post-PAF Groups

No significant differences were found in the length of hospital stay, 30-day mortality, and relapse due to infection recurrence. However, the rate of treatment failure significantly decreased after the implementation of PAF (*p* = 0.04; Table 3).

Table 4 lists the factors associated with treatment failure in ESBL-EC bacteremia. The multivariate analysis revealed that treatment failure was significantly associated with PAF.

Additionally, the breakdown of the source of bacteremia was also similar between the treatment failure and no treatment failure groups. The initial usage rate of non-susceptible antibiotics against ESBL-EC did not differ between the treatment failure and no treatment failure groups (43.8% vs. 54.4%, *p* = 0.46). The initial usage rate of carbapenems was higher in the treatment failure group than in the no -treatment failure group (*p* = 0.089; Table 4).

## 4. Discussion

This study compared the clinical outcomes before and after the implementation of PAF in patients with ESBL-EC bacteremia at our hospital. The comparison of process indicators between the pre- and post-PAF groups revealed that the frequency of de-escalation from broad-spectrum antibiotics to CMZ was significantly higher in the post-PAF group than that in the pre-PAF group. This is because the recommendation for de-escalation to CMZ based on the results of drug susceptibility was actively proposed upon the identification of ESBL-EC bacteremia through PAF implementation. Previous studies on the impact of PAF implementation on antibiotic use have reported improvements in process indicators, including reduced use of targeted antibiotics, in the early years (particularly in the first 1–2 years) after PAF implementation [26,27]. PAF can facilitate the appropriate selection of antibiotics for empirical therapy and reduce the use of broad-spectrum antibiotics [28]. Moreover, PAF conducted by a multidisciplinary AST can reduce the time required for the de-escalation of anti-MRSA agents [22]. The semisynthetic cephamycin, CMZ, is a promising candidate for carbapenem-sparing therapy as it remains stable under ESBL hydrolysis and exerts antibiotic activity against ESBL-EC [29,30]. In addition, previous studies have shown that over 90% of ESBL-producing pathogens, including ESBL-EC and ESBL-producing *Klebsiella pneumoniae*, are susceptible to CMZ [31,32]. The susceptibility rate of CMZ against ESBL-EC in our hospital was satisfactory, with 98.4% (61/62) of cases exhibiting ESBL-EC susceptibility to CMZ over the study period. Although no significant difference was observed in the overall treatment period between the pre- and post-PAF groups, the duration of treatment with broad-spectrum antibiotics in the post-PAF group was shorter than in the pre-PAF group. The proportion of patients switching from intravenous to oral antibiotics did not change after PAF implementation. This may be because no active proposals have been submitted for switching to oral antibiotics, given the lack of evidence for their efficacy against ESBL-EC bacteremia. Nakakura et al. examined 87 cases of ESBL-producing Gram-negative bacteremia and reported no difference in the 30-day mortality between treatment periods of ≤10 days and ≥11 days. However, treatment periods of ≤10 days were found to be associated with the risk of recurrent bacteremia [33]. No differences were observed in the clinical outcomes between short- and long-term treatment for *Enterobacteriaceae* bacteremia [34,35]. Therefore, we did not actively recommend short-term treatment after PAF implementation. The results of the present study indicate that PAF implementation facilitates the appropriate use of antibiotics, including broad-spectrum ones.

No differences were found in the length of hospital stay and duration of antibiotics treatment between the pre-PAF and post-PAF groups. However, the rate of treatment failure in the pre-PAF group was higher than that in the post-PAF group. Inappropriate initial treatment for patients with acute cholangitis can increase the 30-day mortality [36]. In the present study, the initial usage rate of non-susceptible antibiotics against ESBL-EC did not differ between the treatment failure and no treatment failure groups. These results indicate that initial treatment with an inappropriate antibiotic against ESBL-EC has a limited impact on clinical outcomes. However, the initial usage rate of carbapenems was higher in the treatment failure group. This difference may be attributed to the high number of severely ill patients in the treatment failure group, which necessitated the selection of a carbapenem as the first therapeutic intervention. Therefore, the patient’s severity of illness, not the initial choice of antibiotics, directly impacts the clinical outcomes. This may be because the PBS in the treatment failure group tended to be higher than that in the no treatment failure group (Table 4). Nevertheless, no significant differences in laboratory findings or CCI were observed between the two groups, although CCI has been linked to the recurrence of bacteremia [37,38]. Thus, PAF implementation improves not only process indicators but also clinical outcomes, specifically the reduction of treatment failure rates.

This study has several limitations. First, this was a single-center retrospective study with a limited number of cases. Second, BCID2 was introduced at our hospital in December 2022 after PAF implementation. Because BCID2 enables the early diagnosis of ESBL-EC bacteremia, the timing of the introduction of BCID2 may have influenced the de-escalation rate from broad-spectrum antibiotics to CMZ, which was higher in the BCID2 group than in the non-BCID2 group (Appendix A, *p* = 0.023). Moreover, multivariate logistic regression analysis was performed for factors associated with de-escalation from broad-spectrum antibiotics to CMZ in univariate analysis (*p* < 0.05). The multivariate analysis revealed that de-escalation from broad-spectrum antibiotics to CMZ was significantly associated with PAF (Appendix A, *p* = 0.017). This indicates that the influence of BCID2 on de-escalation to CMZ cannot be ruled out and that PAF exerted the greatest influence on de-escalation to CMZ. Third, prior to the introduction of BCID2, ESBL-EC genes could not be quantified. ESBL-EC strains isolated in Japan are mainly *blaCTX-M* strains, which are highly susceptible to CMZ [39,40,41]. The susceptibility of CMZ to ESBL-EC in our hospital is favorable, and it is unlikely that the inability to identify the ESBL gene prior to the introduction of BCID2 affected the results. Finally, because cases with multiple bacteria detected in blood cultures were also included in the analysis, bacteria other than ESBL-EC may have affected the clinical outcomes and process indicators.

Despite these limitations, we believe that our findings are significant because only a few reports have examined the impact of PAF implementation on clinical outcomes in patients with ESBL-EC bacteremia. The findings of our study demonstrate that PAF implementation contributes to the appropriate utilization of antibiotics and improved clinical outcomes in patients with ESBL-EC bacteremia. We hope that this study will encourage the implementation of PAF in more facilities to instigate a collective effort to reduce the incidence of AMR.

## 5. Conclusions

The present study showed that the implementation of PAF increased the de-escalation rate from broad-spectrum to narrow-spectrum antibiotics and reduced the treatment failure rate. Because this study is a single-center retrospective study with a small number of cases, a larger-scale prospective study is required in the future to validate the current findings.

## Figures and Tables

**Figure 1 microorganisms-12-02275-f001:**
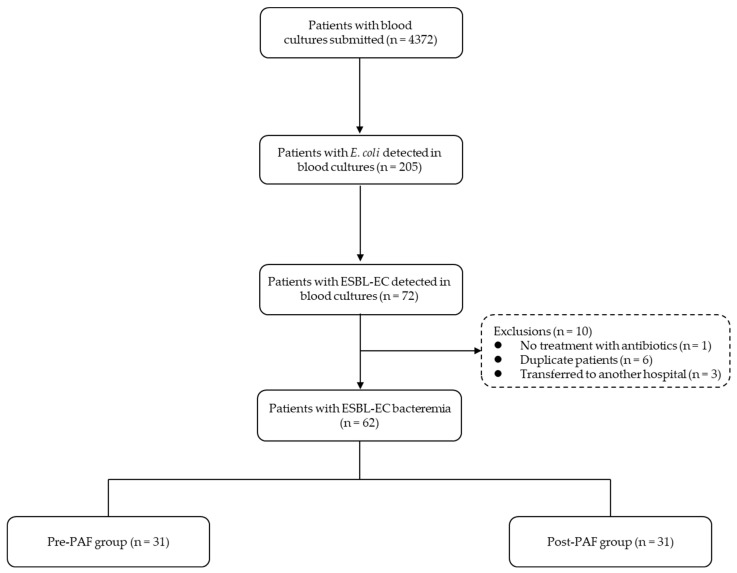
Flow diagram for patient selection with extended-spectrum *β*-lactamase-producing *Escherichia coli* (ESBL-EC) bacteremia before (pre-PAF) and after (post-PAF) implementing prospective audit and feedback (PAF).

**Table 1 microorganisms-12-02275-t001:** Patient characteristics.

Variables		Pre-PAF Group (n = 31)	Post-PAF Group (n = 31)	*p* Value
Age, years ^a^		84 (79–90)	84 (77–89)	0.92
Male sex		12 (38.7)	16 (51.6)	0.31
History of catheter insertion		19 (61.3)	21 (67.7)	0.60
Use of anticancer and immunosuppressive agents		4 (12.9)	7 (22.6)	0.51
Surgical history		9 (29.0)	7 (22.6)	0.56
Pitt bacteremia score ^a^		1 (0–4)	2 (0–2)	0.61
Pitt bacteremia score ≥ 2		14 (45.2)	18 (58.1)	0.30
Charlson comorbidity index ^a^		3 (2–5)	2 (1–4)	0.40
Charlson comorbidity index ≥ 3		19 (61.3)	14 (45.2)	0.20
White blood cell counts, /µL ^a^		9400 (4200–13,300)	10,500 (6300–13,400)	0.42
C-reactive protein, mg/dL ^a^		9.3 (4.7–13.7)	11.1 (5.8–20.4)	0.20
Creatinine, mg/dL ^a^		1.06 (0.66–1.57)	0.96 (0.67–1.52)	0.98
Detection of multiple bacteria in blood culture		4 (12.9)	3 (9.7)	1.00
Sources of bacteremia				0.59
	Urinary tract infection	15 (48.4)	18 (58.1)	
	Biliary tract infection	10 (32.2)	8 (25.8)	
	Respiratory tract infection	4 (12.9)	3 (9.7)	
	Catheter-related bloodstream infection	1 (3.2)	0 (0)	
	Skin and soft tissue infection	1 (3.2)	0 (0)	
	Pancreatic infection	0 (0)	1 (3.2)	
	Unknown	0 (0)	1 (3.2)	
Initial treatment with antibiotics				0.58
	Sulbactam/cefoperazone	8 (25.8)	7 (22.6)	
	Ceftriaxone	8 (25.8)	7 (22.6)	
	Meropenem	6 (19.4)	9 (29.0)	
	Tazobactam/piperacillin	2 (6.5)	4 (12.9)	
	Cefmetazole	2 (6.5)	3 (9.7)	
	Cefotiam	1 (3.2)	1 (3.2)	
	Sulbactam/ampicillin	2 (6.5)	0 (0)	
	Biapenem	2 (6.5)	0 (0)	
Initial treatment with carbapenem		8 (25.8)	9 (29.0)	0.78

Data are expressed as number (%). ^a^ median (interquartile range). PAF: prospective audit and feedback.

**Table 2 microorganisms-12-02275-t002:** Comparison of process indicators between the pre-PAF and post-PAF groups.

Variables	Pre-PAF Group (n = 31)	Post-PAF Group (n = 31)	*p* Value
Blood culture identification 2 panel	0 (0)	22 (71.0)	<0.0001
Number of days from blood culture submission to a positive blood culture result ^a^	1 (1–3)	1 (1–2)	0.74
De-escalation from broad-spectrum antibiotics to cefmetazole	11 (35.4)	25 (80.7)	0.0003
Length of treatment with antibiotics, days ^a^	11 (7–16)	12 (8–16)	0.55
Length of treatment with broad-spectrum antibiotics, days ^a^	7 (3–12.5)	5 (2–8)	0.29
Switching from intravenous to oral antibiotics	3 (9.7)	2 (6.5)	1.00

Data are expressed as number (%). ^a^ median (interquartile range). PAF: prospective audit and feedback.

**Table 3 microorganisms-12-02275-t003:** Comparison of clinical outcomes between the pre-PAF and post-PAF groups.

Variables	Pre-PAF Group (n = 31)	Post-PAF Group (n = 31)	*p* Value
Length of hospital stay, days ^a^	13 (10–22)	15 (11–22)	0.51
Treatment failure	12 (38.7)	4 (12.9)	0.04
30-day mortality	5 (16.1)	1 (3.2)	0.19
Relapse of infections	7 (22.6)	3 (9.7)	0.3

Data are expressed as number (%). ^a^ median (interquartile range). PAF: prospective audit and feedback.

**Table 4 microorganisms-12-02275-t004:** Factors associated with treatment failure (univariate and multivariate analyses).

Variables		No Treatment Failure (n = 46)	Treatment Failure (n = 16)	*p* Value	Multivariate Analysis		*p* Value
					OR	95% CI	
Age, years ^a^		83.5 (76.8–89)	87 (80–92.5)	0.16	—	—	—
Male sex		23 (50.0)	5 (31.3)	0.19	—	—	—
History of catheter insertion		31 (67.4)	9 (56.3)	0.42	—	—	—
Use of anticancer and immunosuppressive agents		8 (17.4)	3 (18.8)	1.00	—	—	—
Surgical history		10 (21.7)	6 (37.5)	0.21	—	—	—
Pitt bacteremia score ^a^		1 (0–2)	2.5 (0.3–6.8)	0.11	—	—	—
Pitt bacteremia score ≥ 2		22 (47.8)	10 (62.5)	0.31	—	—	—
Charlson comorbidity index, median ^a^		3 (2–4)	2 (1–5.8)	0.73	—	—	—
Charlson comorbidity in-dex ≥ 3		26 (56.5)	7 (43.8)	0.37	—	—	—
White blood cell counts, /µL ^a^		10,500 (5875–13,300)	9750 (6225–17,200)	0.89	—	—	—
C-reactive protein, mg/dL ^a^		11 (5.2–17.1)	8.4 (3–16.2)	0.38	—	—	—
Creatinine, mg/dL ^a^		1 (0.65–1.49)	1.2 (0.79–1.62)	0.77	—	—	—
Detection of multiple bacteria in blood cultures		4 (8.7)	3 (18.8)	0.36	—	—	—
Sources of bacteremia				0.54	—	—	—
	Urinary tract infection	25 (54.4)	8 (50.0)		—	—	—
	Biliary tract infection	12 (26.1)	6 (37.5)		—	—	—
	Respiratory tract infection	6 (13.0)	1 (6.3)		—	—	—
	Catheter-related bloodstream infection	0 (0)	1 (6.3)		—	—	—
	Skin and soft tissue infection	1 (2.2)	0 (0)		—	—	—
	Pancreatic infection	1 (2.2)	0 (0)		—	—	—
	Unknown	1 (2.2)	0 (0)		—	—	—
Initial treatment with antibiotics				0.54	—	—	—
	Sulbactam/cefoperazone	12 (26.1)	3 (18.8)		—	—	—
	Ceftriaxone	12 (26.1)	3 (18.8)		—	—	—
	Meropenem	9 (19.6)	6 (37.5)		—	—	—
	Tazobactam/piperacillin	5 (10.9)	1 (6.3)		—	—	—
	Cefmetazole	5 (10.9)	0 (0)		—	—	—
	Cefotiam	1 (2.2)	1 (6.3)		—	—	—
	Sulbactam/ampicillin	1 (2.2)	1 (6.3)		—	—	—
	Biapenem	1 (2.2)	1 (6.3)		—	—	—
Initial treatment with carbapenem		10 (21.7)	7 (43.8)	0.089	3.4	0.9–12.8	0.068
Prospective audit and feedback		27 (58.7)	4 (25.0)	0.04	0.2	0.05–0.78	0.02

Data are expressed as number (%). ^a^ median (interquartile range). OR: odds ratio, CI: confidence interval.

## Data Availability

The original contributions presented in the study are included in the article/Appendix A, further inquiries can be directed to the corresponding author.

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
