# Peer review of "Effects of Prospective Audit and Feedback in Patients with Extended-Spectrum β-Lactamase-Producing Escherichia coli Bacteremia"

_microorganisms, 2024, doi:10.3390/microorganisms12112275_

Round 1

Reviewer 1 Report

Comments and Suggestions for Authors

The manuscript by Yota Yamada and co-authors titled "Effects of prospective audit and feedback in patients with extended-spectrum β-lactamase–producing Escherichia coli bacteremia" is dedicated to the influence of prospective audit and feedback in patients with ESBL–producing E. coli bacteremia. The manuscript is well-written and translated. I did not find any major flaws.

However, there are a few minor issues listed below:

Lines 37 and 196: Perhaps "Enterobacteriaceae" should be italicized.

Line 58: Could you add references to a "few studies"?

Line 69: Could the exclusion of this group bias the findings of this study?

Line 220: Reference # 40 is missing from the reference list.

Additionally, since the study involves patients, it is necessary to add the protocol number from the Ethics Committee meeting.

Line 368: It seems that something has shifted, and the numbering of references is incorrect. Please correct this.

Reviewer 2 Report

Comments and Suggestions for Authors

Yamada Y. et al wrote a paper regarding the impact of prospective audits and feedback (PAF) in antimicrobial stewardship programs. The topic covered is interesting and clearly stated. 

Despite this there are some serious limitations, admitted by the authors.

Major reviews

- Having conducted rapid BCID2 identification only on PAF patients makes the results on the switch to cefmetazole difficult to evaluate. I would therefore be much more cautious in assessing its impact.

- the lack of a 1:1 match makes it difficult to assess whether the two populations pre and post PAF are comparable. In this regard, it would be indicated to add as many characteristics as possible in Table 1. For example, comparison of patients with CCI>3 in the two groups.

Minor revisions

- in all tables it should be specified item by item whether we are talking about median (IQR) or numerosity (%)

- line 233: I would write “the present study seems to suggest...”

- English can be improved. For example line 220 "the sensitivity of ... is high..." 

Round 2

Reviewer 2 Report

Comments and Suggestions for Authors

Revisions made are sufficient to accept the article without further editing.